# The Microbiota and Cancer Cachexia

**DOI:** 10.3390/ijms20246267

**Published:** 2019-12-12

**Authors:** Kelly M. Herremans, Andrea N. Riner, Miles E. Cameron, Jose G. Trevino

**Affiliations:** Department of Surgery, University of Florida College of Medicine, 1600 SW Archer Rd, Room 6165, PO Box 100109, Gainesville, FL 32610, USA; Kelly.Herremans@surgery.ufl.edu (K.M.H.); Andrea.Riner@surgery.ufl.edu (A.N.R.); milesecameron@ufl.edu (M.E.C.)

**Keywords:** cancer cachexia, microbiota, systemic inflammation, gut barrier dysfunction, muscle wasting, dysbiosis, probiotics, prebiotics, fecal microbiota transplantation

## Abstract

Cancer cachexia is a multifactorial syndrome defined by weight loss, muscle wasting, and systemic inflammation. It affects the majority of patients with advanced cancer and is associated with poor treatment response, early mortality and decreased quality of life. The microbiota has been implicated in cancer cachexia through pathways of systemic inflammation, gut barrier dysfunction and muscle wasting. The imbalance of the microbiota, known as dysbiosis, has been shown to influence cancer cachexia. Bacteria that play beneficial and detrimental roles in the disease pathogenesis have been identified. The phenotype of cancer cachexia is associated with decreased levels of *Lactobacillales* and increased levels of *Enterobacteriaceae* and *Parabacteroides*. Currently, there are no treatment options that demonstrate increased survival or the quality of life in patients suffering from cancer cachexia. Through the manipulation of beneficial bacteria in the gut microbiota, different treatment options have been explored. Prebiotics and probiotics have been shown to improve outcomes in animal models of cachexia. Expounding on this mechanism, fecal microbiota transplant (FMT) holds promise for a future treatment of cancer cachexia. Further research is necessary to address this detrimental disease process and improve the lives of patients suffering from cancer cachexia.

## 1. Introduction

Cachexia is a multifactorial syndrome defined by weight loss greater than 5%, weight loss greater than 2% in those who have a BMI < 20 kg/m^2^ or depletion in skeletal muscle mass [1]. Cachexia is seen in approximately 80% of patients with advanced cancer and has been reported to contribute to 30% of cancer deaths [2]. The clinical phenomenon of cachexia has been identified for centuries, originally documented by Hippocrates as “the flesh is consumed, the shoulders, clavicles, chest and thighs melt away… the illness is fatal” [3]. Modern research has elucidated a complex interplay of cytokines, inflammation and metabolic derangement. Cachexia includes not only weight loss, but also adipose tissue wasting, muscle atrophy, and decreased appetite, with metabolic dysfunction preceding these physical signs [2,4]. Cachexia has been noted in many chronic inflammatory conditions such as acquired immunodeficiency syndrome (AIDS), sepsis, autoimmune disorders, chronic lung disease and cancer. Cancer cachexia is particularly devastating as it is predictive of early mortality, poor response to chemotherapy and can even be a direct cause of death [5]. In addition to quantity, cancer cachexia affects the quality of patients’ lives. Muscle and adipose wasting along with progressive anorexia can be particularly distressing to patients and their family members. Early interventions such as exercise, nutritional supplementation, counseling and medications have been trialed to no avail. A number of different pharmacologic therapies have been tested with a focus on appetite stimulants, anabolic agents and metabolic inhibitors. Short courses of corticosteroids, progesterone analogs and more recently cannabinoids have been used to enhance appetite stimulation in patients with cancer cachexia. Anabolic steroids and recombinant growth hormone have been evaluated for their theoretical mechanism of action in muscle anabolism; however, no statistically significant benefit has been discovered [6]. Metabolic inhibitors have also been under investigation for their proposed benefit in decreasing systemic inflammation. Eicosapentaenoic acid (EPA) has been suggested to decrease IL-6, but studies have not consistently shown improvement. Additionally, TNF-α antibodies have not shown any statistically significant benefit in humans despite their mechanism [7]. Overall, pharmaceutical interventions have not shown a significant survival benefit or an improvement in quality of life in patients with cancer cachexia [4].

Emerging research has highlighted the role of the microbiota and its influence on the pathogenesis of cancer cachexia. By definition, the human microbiota are the bacteria, fungi, protozoa and viruses coexisting within the human body. The microbiota is composed of 100 trillion microorganisms, far outnumbering the cells of the human body. A common misconception, the term “microbiome” is defined as the genetic material that makes up these organisms [8]. Recent advances in DNA sequencing technology have allowed for exploration into the previously unknown and underappreciated world of the microbiome. Bacteria, fungi and viruses that could not previously be recognized with standard culturing techniques are now identifiable and can be separated into their taxonomies using high throughput DNA sequencing. Bacteria and archaea are sequenced based on their 16S rRNA subunit and fungal components are differentiated based on their internal transcribed spacer (ITS), 18S rRNA, or 26S rRNA regions [9] The metabolic activity of the microbiota has been evaluated on a multitude of scales ranging from a single cell to the epidemiology of their hosts. At the level of a single cell, flow cytometry, fluorescent in situ hybridization (FISH), single cell mass spectrometry and other techniques have been used to elucidate cellular enzymatic activity, gene content and growth rate. These techniques combined with metagenomics (study of genomes within the microbial community), metatranscriptomics (study of gene expression within the microbial community) and metabolomics (study of small molecules released by a microbial community) allow for novel understandings of microbial metabolism and environmental interactions [10].

Through symbiosis, the microbiota and host have evolved to become a “super-organism” with intertwining processes of nutrition and metabolism [8]. Nearly all of the microbiota reside within the alimentary tract though it has been shown to influence the human body both locally and systemically. Beyond the gastrointestinal tract, associations exist between the oral, skin, respiratory and genitourinary microbiota and the pathogenesis of malignancies [11,12,13,14,15,16,17]. Furthermore, the microbiota has been shown to be intimately involved with the modulation of cancer treatment. In their review, Alexander et al. proposed the “TIMER” mechanistic framework to detail the influence of the microbiota on cancer treatment modalities [18]. This acronym describes the Translocation, Immunomodulation, Metabolism, Enzymatic degradation, Reduced diversity and ecological variation. Interactions between the microbiota and cancer therapies have been proven in multiple human, animal and in vitro studies and provide an exciting new prospect for personalized medicine [18]. Alternatively, the progression of cancer has been shown to alter the makeup of the microbiota. New studies have shown that chemotherapy induced mucositis has been proven to directly influence taxonomic shifts of bacteria [19]. Current research has focused on the influence of the microbiota in carcinogenesis. However, few studies exist on the role of the microbiota in cancer cachexia. This review centers on the current literature available on the influence and interactions of the microbiota and cancer cachexia. It discusses the mechanisms of the microbiota implicated in cancer cachexia and dysbiosis (Figure 1). It further highlights the positive outcomes of restoring beneficial bacterial flora and potential targets for the treatment of cancer cachexia.

## 2. Mechanistic Interplay of the Microbiota in Cancer Cachexia

### 2.1. Systemic Inflammation

Cancer cachexia is a multifaceted clinical syndrome that is characterized by disequilibrium of inflammation, gut permeability and muscle wasting [13,20]. Systemic inflammation is a hallmark of cancer-associated cachexia and when compared to controls, cachectic subjects have a higher level of inflammatory markers. In patients with cancer cachexia, elevations of acute phase reactants, such as C-reactive protein (CRP), fibrinogen and inversely albumin, have been correlated with disease progression, decreased survival and poor quality of life [21]. Pro-inflammatory cytokines have been implicated in a broad spectrum of cancer types, not defined by a single organ. Causality of cachexia has been established in interleukin (IL)-6, IL-1 and tumor necrosis factor (TNF)-α when injected systemically in animal models. TNF-α has been shown to directly cause muscle breakdown through the induction of the ubiquitin-proteasome system (UPS). It causes metabolic derangement through proteolysis and decreased synthesis of protein and lipids [22]. IL-6 has also emerged as an influential cytokine in the pathogenesis of cancer cachexia. IL-6 is multifunctional and is involved in wound healing and tissue regeneration. Conversely, IL-6 has been correlated with tumorigenesis, muscle wasting and decreased survival time in patients with cancer cachexia [23]. Patients with advanced cancer have elevated levels of IL-6 which correlate to weight loss, anemia and depression. Recent trials with toculizumab, a humanized IL-6 receptor antibody, have shown promise in reducing muscle atrophy in recent trials, but the overall safety of this drug continues to require further investigation [24].

Microbiota and systemic inflammation are linked. The role of the microbiota and its influence on inflammation has been identified in a number of disease states including obesity, insulin resistance, cardiovascular disease, inflammatory bowel disease, asthma and carcinogenesis. Recent studies have focused on obesity and the concept of “metainflammation”, which refers to the concept of chronic systemic inflammation and metabolic dysfunction [25]. Cani et al. described that mice with high-fat diets had elevated levels of intestinal Firmicutes and Proteobacteria [26]. They exhibited increased intestinal permeability, greater lipopolysaccharide (LPS) serum concentration and endotoxemia [26]. Systemic inflammation occurred through the activation of Toll-like receptors (TLR) and an overproduction of IL-1β, IL-6, IL-8 and TNF-α [27]. Carvalho et al. aimed to evaluate the affect of antibiotic treatment on subclinical inflammation and its clinical manifestations [28]. In their study design, they used mice fed with high-fat diets and subjected one arm to treatment with antibiotics (ampicillin, neomycin and metronidazole for 8 weeks). Then, they analyzed mouse feces, blood samples and biopsies of liver, adipose and muscle tissue. Using metagenome analysis, they found significant alterations in the microbiota with a reduction in *Bacteriodetes* and *Firmicutes*. Additionally, mice treated with antibiotics showed decreased levels of circulating LPS, decreased levels of fasting glucose, insulin, TNF-α and IL-6 [28]. The link between the human microbiota, systemic inflammation and cancer cachexia has yet to be fully elucidated in current research. Further studies are needed to identify the pathways and mechanisms involved in the pathogenesis and clinical presentation of cancer cachexia. The focus of future research should aim to relate the alterations in the microbiota and systemic inflammation in patients with cancer cachexia.

### 2.2. Gut Barrier Dysfunction

The lining of the gastrointestinal tract serves as a barrier between the internal milieu and luminal contents, which includes the bacteria, fungi, protozoa and viruses that make up the microbiota. It serves to protect the body from intraluminal pathogens through the recognition of foreign microbes and secretion of antibodies, antimicrobial peptides and mucus [29]. Its defense is threefold. It includes a biologic barrier made up of microbiota, an immune barrier made of gastrointestinal immune cells and the mechanical barrier between the intestinal epithelial cells and capillary endothelial cells [30]. Gut barrier permeability of the small and large intestine is regulated by cell junctions (tight junctions, adherens junctions and desmosomes) that connect epithelial cells. Tight junctions, in particular, have been shown to have a dynamic response to a variety of factors, both extrinsically and intrinsically [31]. External factors such as drugs, cytokines and chemicals affect the permeability of the gut barrier [32,33,34]. More recently attention has been turned to the internal factors influencing gut barrier dysfunction, specifically the effects of the microbiota. Current literature has hypothesized that resident microbiota cause gut barrier dysfunction leading to increased translocation of bacterial toxins and subsequent systemic inflammation. It has been proposed that bacterial endotoxins such as LPS are able to seep through a more permeable gut barrier and into the bloodstream. As noted above, this leads to systemic inflammation and potentially the clinical presentation of cancer cachexia [35].

Puppa et al. demonstrated evidence of this correlation in their study of Apc^Min/+^ cachexia animal model. The Apc^Min/+^ mouse is an animal model with a nonsense point mutation in the tumor suppressor gene *Apc* that causes colonic tumor development. It is used as a model for cancer cachexia as mice begin to exhibit progressive weight loss when tumor burden progresses. Gut barrier dysfunction was measured by evaluating permeability to neutral hydrophilic polymers. When compared to controls, Apc^Min/+^ mice demonstrated increased gut barrier permeability. The timing of onset of gut barrier dysfunction correlated with both the onset and progression of cancer cachexia. Serum concentrations of IL-6 also increased in parallel with worsening cachexia and gut barrier permeability. Measurements of endotoxemia were fivefold higher in severely cachectic mice. Authors further described a number of major shifts in the metabolism of Apc^Min/+^ mice, including sequelae of hypothermia, hypertriglyceridemia and insulin resistance. These metabolic changes are also frequently noted in patients with progressive cachexia [36].

Jiang et al. performed a comparative study of human patients with gastric adenocarcinoma to further evaluate the relationship between cancer cachexia, microbiota, intestinal barrier dysfunction, bacterial translocation and systemic inflammation [37]. They evaluated the changes in microbial contents, structural basis of tight junctions, and inflammatory cytokines. They studied the differences between cachectic patients and noncachectic patients using a unique population of patients with gastric cancer involving the transverse mesocolon. Cachectic patients were defined using clinical criteria of weight loss >10% of preillness weight and CRP >10 mg/L. They used a “sugar-drink test” to measure the gradient in urine as a marker for intestinal permeability. Intraoperative samples of the wall of the large intestine, mesenteric lymph nodes and blood samples from the middle colic, portal and peripheral veins were obtained. Fecal samples were obtained prior to any intervention. Their results indicated that cachectic patients with gastric adenocarcinoma had higher levels of intestinal barrier dysfunction with increased levels of claudin (channel-forming) transmembrane proteins and decreased levels of occludin transmembrane proteins. Cachectic patients exhibited a higher level of bacterial translocation, increased systemic inflammatory cytokines and significant differences in the diversity of intestinal flora, although individual bacterial species were not identified [37].

Bindels et al. further evaluated the interactions between cancer cachexia, gut barrier dysfunction and the microbiome through both human and animal studies [38]. Their mouse model for cancer cachexia was generated by the ectopic transplantation of C26 colon carcinoma cells. Using control mice, they evaluated the pathologic changes occurring in cancer cachexia. They found significant evidence of alterations in intestinal homeostasis in C26 cachectic mice. This is best demonstrated by an overall decrease in intestinal tissue weight, increased villi length and crypt depth and increased gut permeability with increased claudin proteins. Microbial composition was also altered with an increase in *Enterobacteriaceae*. Increased gut permeability has been associated with an increase in proinflammatory bacterial translocation. The acute phase reactant lipopolysaccharide-binding protein (LBP) was used as a marker for translocation and increased exogenous antigen load. This study showed an increase in both IL-6 and LBP correlating to the manifestation of cancer cachexia. In search of the underlying mechanisms to gut barrier dysfunction, they pair-fed control mice to evaluate the influence of anorexia. They found that intestinal barrier dysfunction and microbial changes in C26 mice could not be attributed to anorexia. Interestingly, they administered an anti-IL-6 antibody and evaluated the downstream effects. In the C26 model, it not only prevented alterations in the microbiota but also improved weight loss, muscle atrophy and food intake. However, there was a slight increase in tumor size. To evaluate this existing data in humans, a prospective cross-sectional study on patients with lung and colon cancer was performed. Serum measurements of IL-6 and LBP were obtained in cachectic and noncachectic patients. They found that cachectic patients demonstrated elevations in serum IL-6 and LBP. Additional multivariate analysis revealed that increased LBP was significantly predictive of poor outcome in cancer patients. It was proposed that in the future LBP might be utilized as a biomarker in cancer cachexia [38]. Given this data, it is conceivable that the immune dysfunction may be both a cause and an effect of intestinal dysfunction and subsequent bacterial translocation. Further research is needed to elucidate the intricacies of the vicious cycle of gut barrier dysfunction, dysbiosis, bacterial translocation and systemic inflammation.

### 2.3. Muscle Wasting

The loss of skeletal muscle is a key factor in the development of cancer cachexia. A decrease in muscle mass is associated with a loss in independence and overall quality of life. The main function of muscle tissue is skeletal stabilization, but it also plays a role in macronutrient storage and the excretion of cytokines. Recent studies have examined the endocrine nature of skeletal muscle and the cross-talk between other organ systems. As current literature continues to evolve, focus has shifted toward exploring the notion of a gut–muscle axis.

Recently, Nay et al. evaluated the interactions between the gut microbiota and muscle function [39]. They exposed mice to 21 days of broad-spectrum antibiotics and studied how it altered skeletal muscle function. They found that mice treated with antibiotics had decreased endurance and increased muscle fatigue. Mice resumed their function after natural bacterial reseeding. Notably, these changes were not associated with changes in muscle mass, muscle composition or mitochondrial function. However, the authors found a parallel relationship to G protein-coupled receptor 41, sodium glucose cotransporter 1 and muscle glycogen levels. These results support an interaction between the gut microbiota, glucose metabolism/storage and muscle function [39].

Frost et al. explored the influence of LPS on cytokine stimulation in skeletal muscle [40]. They showed that LPS modulates the secretion of inflammatory cytokines, specifically TNF-α and IL-6, from muscle cells both in vitro and in vivo. Using C2C12 myoblasts, they monitored the cytokine response from within these cell types. They found that administration of LPS resulted in an increase of TNF-α and IL-6 from myoblasts. These results were blunted in mice with a TLR-4 mutation, suggesting its involvement in this process [40]. Additional studies have exhibited a correlation between circulating IL-6 and the suppression of protein synthesis and muscular atrophy. Haddad et al. evaluated the direct effects of IL-6 on muscle [41]. Administration of IL-6 resulted in overall muscular atrophy and specifically loss of myofibrillar protein. They proposed that this consequence occurred due to downregulation of growth factor-mediated intracellular signaling [41]. TNF-α has also been shown to accelerate apoptosis in muscle cells. Li et al. evaluated the effect of TNF-α on muscle cells both in vitro and in vivo [42]. They found that TNF-α upregulates atrogin1/MAFbx gene expression, which increases muscle wasting. This effect was noted to take effect within 2 h in C2C12 myotubes and within 4 h in mouse skeletal muscle [42]. Put into context with the aforementioned data regarding gut barrier dysfunction and systemic inflammation, the direct activation of muscular and systemic cytokines by LPS may be another step to further explain the interworking of the gut microbiota–muscle axis.

## 3. Dysbiosis

Diversity in the gut microbiota has become increasingly recognized within the last ten years. The popularity of this topic has intensified as investigators continue to discover new applications of the field. The microbiota is not a stagnant physiologic feature, but a dynamic aspect of the human body. Alterations of microbial composition have been shown to occur primarily through the first three years of life and then again later in life, after the age of 65 [43]. Throughout the human lifespan, the microbiota is primarily comprised of the *Bacteroidetes* and *Firmicutes* bacterial species [44]. The Human Microbiome Project is currently working to sequence the microbial genome and understand the role of the microbiome in health and human disease. Their goal is to determine whether humans share a core microbiome and to evaluate how deviation for the norm correlates with different disease states [45].

A landmark study by Bäckhed et al. first identified how dysbiosis effects host metabolism [46]. They evaluated the response of germ-free mice to the initiation of the Western-style, high-fat diet and concluded that the microbiota influences the host metabolism. Specifically, they found that germ-free mice exhibited elevated skeletal muscle and liver levels of AMP-activated protein kinase (AMPK) and elevated levels of fasting-induced adipose factor. Additional studies have highlighted the involvement of the microbiota in amino acid bioavailability and diversifying the bile acid profile [46].

Bindels et al. have performed several studies over the last decade to further elucidate the role of the microbiota in cancer cachexia. Using a mouse model of leukemia, they were able to define the differences in microbial species between cachectic and noncachectic mice. Cells containing Bcr-Abl ectopic expression were transplanted into mice and they subsequently developed the clinical picture of cancer cachexia. DNA sequencing of the microbiota was then performed using 16s rRNA subunit analysis. When compared to controls, cachectic mice exhibited a fifty-fold decrease in *Lactobacillus* species, particularly in *L. reuteri* and *L. johnsonii/gasseri*. Levels of *Enterobacteriaceae* and *Parabacteroides goldsteinii* were found to be increased [47,48].

In a separate study, they further detailed the deleterious effects of *Enterobacteriacea* and specified *Klebsiella oxytoca* as a gut pathobiont. A pathobiont is defined as a potentially pathogenic bacterium that is not harmful in normal settings. They sought to answer the question why cachectic mice exhibited a lower colonization resistance to *K. oxytoca* and identified the role of host-derived nitrate in the reduction of PPAR-γ signaling. This perpetuates the growth of *K. oxytoca* [49]. Evaluation of dysbiosis has not been limited to the gut. Li et al. analyzed the skin flora microbiota in patients with cancer cachexia and compared it to normal controls [14]. Though confounding factors may be present, they found that patients with cancer cachexia had significantly reduced *Corynebacterium* species in their skin microbiota when compared to healthy patients [14]. Overall, identification of microbial disturbances in patients suffering from cancer cachexia allows for further targeted research. In combination with the DNA sequencing collection of the human microbiome project, potential for new therapeutic opportunities exist in the treatment of cancer cachexia.

## 4. Therapeutics

### 4.1. Probiotics

Probiotics have become an increasingly popular research focus in the last three decades. Defined as “live microorganisms that, when administered in adequate amounts, confer a health benefit on the host,” probiotics have been found to have a multitude of effects both locally and systemically in the human body [50]. The study of probiotics and cancer cachexia is still in its infancy but intriguing new evidence has been proposed.

Varian et al. recently evaluated the role of *L. reuteri* as a probiotic in cancer cachexia [51]. Using the widely accepted Apc^MIN/+^ mouse model of colon cancer cachexia, they administered *L. reuteri* via drinking water. They found that administration of this lactic-acid Gram-positive bacterium was associated with larger gastrocnemius muscle mass and decreased evidence of muscle atrophy. Oral *L. reuteri* administration was also associated with an increased lifespan, larger thymus and a decrease in *FoxN1* expression, a transcription factor involved in systemic inflammation [51]. Bindels et al. has led the field in probiotic trials in patients with cancer cachexia [47]. In the aforementioned study, they noted a decrease in the levels of *L. reuteri* and *L. johnsonii/gasseri* in a leukemia mouse model of cachexia. Upon supplementation with oral *L. reuteri*, these mice exhibited reduced expression of muscle atrophy markers, particularly in the gastrocnemius and tibialis muscles. They also demonstrated a reduction in systemic inflammatory cytokines (IL-6, monocyte chemoattractant protein-1, IL-4, granulocyte colony-stimulating factor) [47]. In additional studies, they continued to elaborate on this evidence by utilizing a symbiotic approach using both a probiotic (*L. reuteri*) and prebiotic (inulin-type fructans). They evaluated the effect of this combination in both colon cancer (C26) and leukemic (BaF) mouse models of cancer cachexia. They found that administration of this combination was associated with decreased cancer cell proliferation and muscle wasting. Furthermore, mice treated with *L. reuteri* and inulin-type fructans showed a decreased morbidity and prolonged survival [48]. The study of probiotics continues to proliferate as it hold promise to become a new treatment modality for patients with cancer cachexia. By way of many new emerging medical treatments, the safety of probiotics in cancer patients will need to be thoroughly evaluated.

As this new area of research continues to proliferate, the safety of probiotics in cancer patients has been examined. Redman et al. performed a systematic review of seventeen randomized controlled studies involving the safety of probiotics in cancer patients [52]. They reported a large variety of adverse effects, including an increased risk of sepsis in patients with malignancies. Given the patients’ predisposition for adverse effects from malignant progression, there has been insufficient evidence to support direct causality [52]. Additional studies are needed to evaluate the safety and efficacy of probiotics in patients with cancer cachexia prior to their clinical implementation

### 4.2. Prebiotics

Many different avenues of altering the gut microbiota have been explored. In this pursuit, the idea of a prebiotic was developed. Prebiotics are described as a “nondigestible food ingredient that beneficially affects the host by selectively stimulating the growth and/or activity of one or a limited number of bacteria in the colon, and thus improves host health.” Bindels et al. specifically evaluated the influence of pectic oligosaccharides (POS) and inulin (INU) on the effect of cancer cell proliferation in their leukemic mouse model of cachexia [53]. In comparing the two prebiotics, they found that POS decreased metabolic alterations, delayed anorexia and reduced fat mass loss. However, it did not affect the rate of hepatic cancer cell invasion. Conversely, the use of INU in this mouse model exhibited a decrease in hepatic cancer cell invasion [53]. Huang et al. evaluated this concept using the Apc^MIN/+^ mouse model of colon cancer [54]. They administered triterpene saponins (specifically ginsenoside-Rb3 and ginsenoside-Rd) orally to mice and evaluated the downstream effect on gut epithelium, inflammatory markers and microbiota. Though they did not specifically evaluate the changes in body weight, they reported that mice receiving ginsenoside-Rb3 and ginsenoside-Rd had improvement in their gut epithelium and a decrease in pro-inflammatory markers and cachexia-associated bacteria [54]. Though this research is promising, there is a paucity of human data and further studies are needed to make definitive conclusions.

### 4.3. Fecal Transplantation

Fecal microbiota transplantation (FMT) is defined as “the engraftment of microbiota from a healthy donor into a recipient, which results in restoration of the normal gut microbial community structure.” The technology of FMT has existed for over 50 years, but recent research in this therapy has increased exponentially [55]. The majority of studies have focused on the treatment utility for refractory *Clostridium difficile* infection and historically, FMT has been administered using fecal enemas, nasoduodenal tubes or colonoscopy. The safety of this process has recently been called into question, particularly in immunocompromised patients. Adverse events following FMT are reported to be 28.5% with the majority of complications associated with abdominal pain and discomfort. Serious adverse reactions such as infection, peritonitis and disease relapse are not uncommon at 9.2%. It is important to note that the risk of death was estimated to be 3.5% [56]. However, when compared with FMT in immunocompentent patients, there was no statistical difference in adverse events and this study was likely confounded by the original indications for FMT. Wang et al. performed a systematic review that analyzed 50 publications and a total of 1089 immunocompromised patients that received FMT [57]. This involved a wide variety of patients, including those with HIV, AIDS, solid organ transplant and malignancy. They concluded that there appears to be no increased risk of FMT administration in immunocompromised patients, but data on long-term outcomes is not available [57]. Theoretically, FMT may be a safe option in the treatment of cancer cachexia, but additional research is needed in order to evaluate its overall safety and efficacy.

## 5. Conclusions

Cachexia remains a devastating problem for cancer patients. This clinical diagnosis affects their overall treatment course, survival and quality of life. Cancer cachexia is a complex manifestation of systemic inflammation, gut barrier dysfunction and muscle wasting. The initial catalyst of this deadly triad remains unidentified. Within the last two decades, studies have aimed to define new treatment strategies for combating cancer cachexia. Currently, no medical therapy in clinical practice has provided a meaningful survival benefit or improvement in quality of life. Studies on probiotics and prebiotics have been successful in mouse models but remain in their infancy. They hold the promise of providing future treatment options for patients with cancer cachexia. The technology of FMT has been available for a number of years, but research on this treatment modality has recently flourished. Based on its mechanism of restoring the microbial structure, FMT has the possibility to provide a new and innovative treatment strategy for patients with cancer cachexia. Due to the novelty of these therapeutic options, long-term safety profiles have yet to be described in the literature. Further studies on their safety profiles in patients with cancer cachexia will need to be explored. Many unanswered questions exist regarding the interplay of the microbiota and cancer cachexia. Unearthing the additional components of these local and systemic interactions is critical. New solutions for the treatment of this disease process have the potential to prolong survival and ultimately, improve the quality of life for patients suffering from cancer.

## Figures and Tables

**Figure 1 ijms-20-06267-f001:**
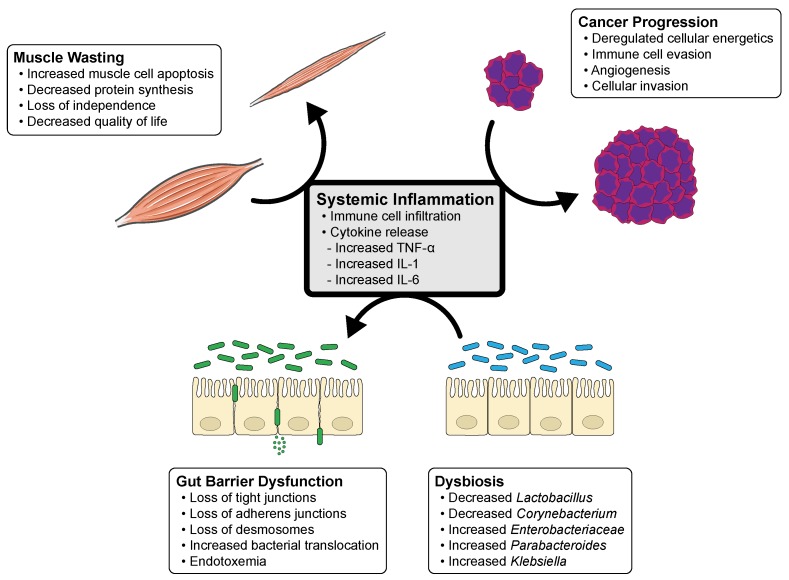
Cycle of systemic inflammation and the influence of cancer progression, dysbiosis, gut barrier dysfunction and muscle wasting in cancer cachexia. This diagram illustrates the interplay of systemic inflammation, likely from disease progression, which leads to dysbiosis and gut barrier dysfunction. Gut barrier dysfunction leads to increasing permeability of pathologic bacteria and endotoxemia, and perpetuates systemic inflammation and muscle wasting.

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
