# Peer review of "The Microbiota and Cancer Cachexia"

_ijms, 2019, doi:10.3390/ijms20246267_

Round 1
Reviewer 1 Report
The review by Herremans and collaborators is a survey of the literature available on a very hot topic, the role of microbiota in the onset and treatment of cachexia, a devastating syndrome that generally occurs in cancer patients and that at present is not treatable. The review is well done, so I will put here few suggestions that could further improve it:
in the last years metabolic alterations have become highly relevant to the pathogenesis of cachexia. While the review very well describes the relationship between microbiota and inflammation, the effect of both cancer-induced dysbiosis and gut barrier leackage on host metabolism are only marginally addressed. As an example, a recent study supports a role for gut microbiota in regulating muscle function with a peculiar link to glucose homeostasis (PMID:31039010); along the same line, cachexia can be significantly exacerbated by anti-cancer treatments. Few sentences addressing the possibility that microbiota modulations can impinge, both in positive and in negative, on treatment side-effects would be welcome; the last chapter is a Conclusion rather than a Discussion; the authors define cachexia as a 'clinical diagnosis' (Abstract and Introduction), which is not correct, it is a multifactorial syndrome; the sentence in lines 75-76 does not sound fine. What do the authors mean with 'uncoupling of the UPS'? similarly, at line 79 the authors talk about IL-6 'pathological state'. What does it mean? what is the 'heterologous' mutation occurring in the ApcMin/+ mouse? line 175 and throughout the text, including Figure 1: the authors talk about 'muscle catabolism' which is not correct. Just tissue components (lipids proteins, etc.) can be degraded, not the tissue as a whole.Author Response
The review by Herremans and collaborators is a survey of the literature available on a very hot topic, the role of microbiota in the onset and treatment of cachexia, a devastating syndrome that generally occurs in cancer patients and that at present is not treatable. The review is well done, so I will put here few suggestions that could further improve it: in the last years metabolic alterations have become highly relevant to the pathogenesis of cachexia. While the review very well describes the relationship between microbiota and inflammation, the effect of both cancer-induced dysbiosis and gut barrier leakage on host metabolism are only marginally addressed. As an example, a recent study supports a role for gut microbiota in regulating muscle function with a peculiar link to glucose homeostasis (PMID:31039010);
The reviewer makes an excellent point. The gut microbiota, host metabolism and cancer cachexia are certainly intertwined. Unfortunately there is a paucity of information that directly evaluates the effects of dysbiosis, gut barrier dysfunction and its associated metabolic effects particularly in humans. I have added in an early study by Bäckhed et al.that shows the effects of dysbiosis on the host metabolism. My goal is to emphasize the multiple ways the micriobiota effects metabolism. In the study by Puppa et al., the metabolic effects of gut barrier dysfunction were briefly described. I have added more detail about the associated metabolic derangements seen in ApcMin/+mice. Additionally, I really enjoyed the suggested article by Nay et al. and have added into my section about the gut microbiota and muscle wasting. Thank you for this interesting addition.
along the same line, cachexia can be significantly exacerbated by anti-cancer treatments. Few sentences addressing the possibility that microbiota modulations can impinge, both in positive and in negative, on treatment side-effects would be welcome;
This is a great idea. I have added in a broad description of the interactions between the mechanisms of interactions between the microbiota and anti-cancer treatments. This is an exciting new topic and certainly deserves attention.
the last chapter is a Conclusion rather than a Discussion;
Thank you for catching that discrepancy. I have adjusted the title as recommended.
the authors define cachexia as a 'clinical diagnosis' (Abstract and Introduction), which is not correct, it is a multifactorial syndrome;
Thank you for identifying this mistake. I have changed the description in both the abstract and introduction.
the sentence in lines 75-76 does not sound fine. What do the authors mean with 'uncoupling of the UPS'?
The reviewer makes a great point. The prior wording did not explain what I was trying to say. Instead, I have changed it to describe the interaction between TNF-α and muscle breakdown through UPS.
similarly, at line 79 the authors talk about IL-6 'pathological state'. What does it mean?
Thank you for noticing this mistake in wording. I have changed this sentence to discuss the negative and positive value of IL-6.
what is the 'heterologous' mutation occurring in the ApcMin/+ mouse?
This is a great question. I have changed the wording to a “nonsense point“ mutation that will be more straightforward for readers.
line 175 and throughout the text, including Figure 1: the authors talk about 'muscle catabolism' which is not correct. Just tissue components (lipids proteins, etc.) can be degraded, not the tissue as a whole.
Thank you for highlighting this much-needed change. I have made adjustments throughout my paper as appropriate.
Reviewer 2 Report
Interesting review, not discussed enough elsewere in the literature. The Manuscript is in general well structured, however minor changes are needed.
Abstract
Line 18: alterations may be made in the disease: please reformulate
Introduction:
Line 33: Are you talking more about cancer associated cachexia anorexia syndrome than about cachexia? Please reevaluate this and adjust when applicable throughout the manuscript
Line 38-43: Please rewrite with adding some references about failed treatment of CAS despite total parenteral nutrition, mention maybe some medications and substances targeting the catabolism of muscle and fat tissue.
Please add about reasons for shifts in Microbiota in Cancer patients
Please add about the diagnostic progress regarding microbiota highlighting the different methods for the examination of bacterial diversity and other examinating the bacterial “metabolic” activity
2.1.:
Are there any studies about link between the human microbiota and systemic inflammation in cancer Cachexia patients?
The figure is good informative, however, i think the compact informations provided in the figure should be better provided at one section of the manuscript, e.g. in the introduction or discussion section.
Author Response
The Microbiota and Cancer Cachexia
IJMS Review Edits
To the editorial office and reviewers,
Thank you for dedicating your time to review this article and adding your thoughtful expertise. Below we feel we have thoroughly addressed the reviewers concerns as outlined in our responses.
Reviewer 2:
Abstract
Line 18: alterations may be made in the disease: please reformulate
This is a great point. The grammar has been changed to better get my idea across.
Introduction:
Line 33: Are you talking more about cancer associated cachexia anorexia syndrome than about cachexia? Please reevaluate this and adjust when applicable throughout the manuscript
Thank you for your comment and observation. In our manuscript we are discussing the role of how the microbiome and muscle interplay cancer cachexia and leads to significant muscle wasting leading to a horrible clinical syndrome. While anorexia, lack of oral intake due to lack of appetite, is a significant problem in cancer patients, clinically muscle wasting is an irreversible phenomena that has been suggested to be independent of nutritional intake. We believe anorexia will be a significant clinical discussion for our group as we begin to tackle ways to stabilize the effects of cancer associated muscle wasting.
Line 38-43: Please rewrite with adding some references about failed treatment of CAS despite total parenteral nutrition, mention maybe some medications and substances targeting the catabolism of muscle and fat tissue.
The reviewer makes an excellent point here. I have added an overview of medications and substances that have been attempted, though mostly unsuccessfully, it patients with cancer cachexia.
Please add about reasons for shifts in Microbiota in Cancer patients
Thank you for this feedback and idea. I have added a section from 86-95 discussing the influence between the microbiota and the progression of cancer as well as the treatment. Unfortunately, there is minimal information of the direct cause of shifts in microbiota in patients with cancer cachexia.
Please add about the diagnostic progress regarding microbiota highlighting the different methods for the examination of bacterial diversity and other examinating the bacterial “metabolic” activity
This was a great addition to the paper. I have added more information from line 73-81 about the techniques used and levels of study within the microbiome.
2.1.:
Are there any studies about link between the human microbiota and systemic inflammation in cancer Cachexia patients?
The reviewer makes an excellent suggestion. The closest study to evaluating the link between the human microbiota, systemic inflammation and cancer cachexia is the comparative study performed by Jiang et. al. In this study, they exam the gut barrier dysfunction and the diversity of the intestinal flora in cachectic patients but do not specify the different strains. This link has been tested in mouse models (Bindel et. al.) but it is an area that would benefit from further research. Many additional studies in humans have linked bacterial strains to carcinogenesis, but not specifically cancer cachexia or systemic inflammation.
The figure is good informative, however, i think the compact informations provided in the figure should be better provided at one section of the manuscript, e.g. in the introduction or discussion section.
Thank you for your feedback. The figure has been edited and moved to after the introduction.
Round 2
Reviewer 1 Report
The authors satisfactorily addressed the points raised in the first revision round